# A Molecular Approach to Understanding the Role of Diet in Cancer-Related Fatigue: Challenges and Future Opportunities

**DOI:** 10.3390/nu14071496

**Published:** 2022-04-02

**Authors:** Sylvia L. Crowder, Mary C. Playdon, Lisa M. Gudenkauf, Jennifer Ose, Biljana Gigic, Leigh Greathouse, Anita R. Peoples, Alix G. Sleight, Heather S. L. Jim, Jane C. Figueiredo

**Affiliations:** 1Department of Health Outcomes and Behavior, Moffitt Cancer Center, Tampa, FL 33617, USA; sylvia.crowder@moffitt.org (S.L.C.); lisa.gudenkauf@moffitt.org (L.M.G.); heather.jim@moffitt.org (H.S.L.J.); 2Cancer Control and Population Sciences, Huntsman Cancer Institute, University of Utah, Salt Lake City, UT 84112, USA; mary.playdon@hci.utah.edu; 3Department of Nutrition and Integrative Physiology, University of Utah, Salt Lake City, UT 84112, USA; 4Department of Population Health Sciences, University of Utah, Salt Lake City, UT 84112, USA; jennie.ose@hci.utah.edu (J.O.); anita.peoples@hci.utah.edu (A.R.P.); 5Huntsman Cancer Institute, Salt Lake City, UT 84112, USA; 6Department of General, Visceral and Transplantation Surgery, Heidelberg University Hospital, 69047 Heidelberg, Germany; biljana.gigic@med.uni-heidelberg.de; 7Human Science and Design, Robbins College of Health and Human Sciences, Baylor University, Waco, TX 76798, USA; leigh_greathouse@baylor.edu; 8Department of Physical Medicine and Rehabilitation, Cedars-Sinai Medical Center, Los Angeles, CA 90048, USA; alixsleight.warner@cshs.org; 9Center for Integrated Research in Cancer and Lifestyle, Cedars-Sinai Medical Center, Los Angeles, CA 90048, USA; 10Department of Medicine, Samuel Oschin Comprehensive Cancer Institute, Cedars-Sinai Medical Center, Los Angeles, CA 90048, USA

**Keywords:** metabolomics, diet-related disease, dietary intervention, cancer survivors, nutrition

## Abstract

Cancer-related fatigue (CRF) is considered one of the most frequent and distressing symptoms for cancer survivors. Despite its high prevalence, factors that predispose, precipitate, and perpetuate CRF are poorly understood. Emerging research focuses on cancer and treatment-related nutritional complications, changes in body composition, and nutritional deficiencies that can compound CRF. Nutritional metabolomics, the novel study of diet-related metabolites in cells, tissues, and biofluids, offers a promising tool to further address these research gaps. In this position paper, we examine CRF risk factors, summarize metabolomics studies of CRF, outline dietary recommendations for the prevention and management of CRF in cancer survivorship, and identify knowledge gaps and challenges in applying nutritional metabolomics to understand dietary contributions to CRF over the cancer survivorship trajectory.

## 1. Introduction

The American Cancer Society defines a cancer survivor as “anyone who has ever been diagnosed with cancer no matter where they are in the course of their disease” [1]. For many survivors, cancer-related fatigue (CRF) is a “distressing, persistent, subjective sense of physical, emotional, and/or cognitive tiredness or exhaustion related to cancer or cancer treatment that is not proportional to recent activity and interferes with usual functioning” [2], and prevention and interventional strategies are needed to address this growing challenge. CRF is considered one of the most frequent side-effects of cancer treatment [3], prompting the National Cancer Institute (NCI) to identify CRF as one of the top five high-priority cancer research areas [4]. CRF negatively impacts work, social relationships, and daily activities, resulting in a significant impairment in quality of life [5,6,7]. Despite these negative implications, CRF is underreported by cancer survivors and physicians [2,8]. A recent systematic review suggests that over half of all cancer survivors experience CRF, [3] and 96% of patients undergoing chemotherapy experience CRF [9]. Clinical symptoms of CRF include cognitive difficulties, hot flashes, functional decline, and depression, all of which result in a significant reduction in quality of life [10,11]. Although there is no evidence-based, gold-standard treatment for CRF, current interventions include education and counseling, skills training in fatigue management strategies, and specific nonpharmacologic and pharmacologic interventions [2]. Although exercise has been shown to reduce CRF [12,13], many patients are unable to exercise consistently [14,15]. Psychosocial interventions demonstrate benefit [16,17,18], although they tend to be time-intensive, limiting uptake, compliance, maintenance, and disseminability. Pharmacologic interventions have not resulted in substantial decreases in CRF. Paroxetine, sertraline, modafinil, and armodafinil have produced no benefit for CRF in randomized trials [19,20,21,22,23,24]. Evidence is mixed regarding the effects of methylphenidate [25,26,27,28] and uptake has been limited, perhaps because methylphenidate can be habit-forming and is associated with side effects including agitation, insomnia, and loss of appetite [29]. CRF is a complex syndrome, and treatment strategies of CRF may differ depending on the survivor’s clinical status (e.g., under active treatment, post-treatment, end-of-life care). While corticosteroids may provide short-term relief for fatigue and improve quality of life in terminal patients, they may not be suitable for long term use in post-treatment survivors [30]. Additional treatment options for CRF are clearly needed. Nutritional interventions offer a promising strategy for reducing CRF [31], but research is needed to elucidate the potential contributions of diet to CRF development and severity to isolate dietary targets that could help mitigate its devastating impacts.

Nutritional metabolomics, the study of diet-related small molecules, is an emerging field of study. It has recently been leveraged to identify objective biomarkers of dietary exposures, and to identify biochemical pathways underlying complex relationships between diet and health or disease. Nutritional metabolomics also encompasses the metabolic state of an individual based on nutritional status (i.e., the state of a person’s health in terms of the nutrients in their diet) [32,33,34]. This is particularly relevant to cancer survivors, as evidence supports the role of nutritional status in the etiology of CRF [31]. Moreover, levels of circulating pro-inflammatory biomarkers are increased in cancer survivors with fatigue [5]. Thus, specific diets aimed at reducing chronic inflammation (e.g., a Mediterranean diet or other specific dietary patterns shown to reduce circulating inflammatory biomarkers [35]) may mitigate CRF via modulating inflammation, and additionally address nutritional deficiencies. Identifying other biochemical pathways linked with CRF could offer additional targets for dietary change. In all, evidence-based nutritional guidelines for alleviating CRF are critically needed, despite the high prevalence of nutritional complications related to fatigue in cancer survivors [36]. In this position paper, we examine CRF risk factors, summarize metabolomics studies of CRF, outline dietary recommendations for the prevention and management of CRF in cancer survivorship, and identify knowledge gaps and challenges in applying nutritional metabolomics to understand dietary contributions to CRF after a cancer diagnosis.

## 2. Risk Factors of Cancer-Related Fatigue

Several barriers currently prevent adequate risk assessment and management of CRF: an incomplete understanding of CRF biological mechanisms; the complexity of CRF risk factors; a paucity of effective CRF treatments beyond existing pharmacological interventions; and the existing interventions being time intensive and difficult to implement [37]. Identifying the biological, genetic, and biobehavioral risk factors for CRF will highlight opportunities to target diet as a tool for CRF prevention and management.

### 2.1. Biological and Genetic Risk Factors

Numerous biological and genetic mechanisms have been linked to CRF. Cancer and its treatment are associated with increased circulating cytokines [38,39], hypothalamic-pituitary-adrenal axis dysfunction [39,40,41], 5-hydroxytryptamine (5-HT) (serotonin) dysfunction [21,39], and altered adenosine triphosphate (ATP) metabolism [39,42], which are hypothesized to contribute to CRF burden. Preliminary longitudinal studies in cancer cohorts support the hypothesis that inflammatory processes involving single nucleotide polymorphisms (SNPs) in inflammation-related genes (e.g., interleukin-1 beta (*IL1ß*), in [43] terleukin-6 (*IL6*), and tumor necrosis factor alpha (*TNFα*)) are associated with increased risk of CRF [44,45,46,47], although the precise mechanisms remain unclear. In a study of long-term survivors of testicular cancer, higher levels of circulating interleukin-1 receptor antagonist (IL-1ra) and C-Reactive protein (CRP) were found in those survivors with chronic CRF, thus lending support to the hypothesis that low-grade inflammatory processed are involved in chronic CRF [43]. Cytokine dysregulation may also contribute to other nutritional risk factors, including anemia, cachexia, anorexia, and infection, all of which further contribute to fatigue [38,48]. Anemia results in decreased oxygen delivery to tissues, resulting in fatigue, dyspnea, and dizziness [49]. Hypoxia-related organ dysfunction is suspected to be related to anemia or hemoglobin dysfunction, which may cause fatigue [39]. Cachexia has been associated with increased levels of interleukins and TNF-α, and abnormalities in energy metabolism in metabolic organs such as the liver, heart, skeletal muscle, adipose tissue, and the gastrointestinal tract [9,50]. Physical conditions such as anorexia, malnutrition, or infections may interact with affective symptoms (e.g., chemical dependency, irritability, and feelings of hopelessness), further contributing to CRF [50]. Other inflammatory processes linked to CRF include cellular immunity marked by leukocyte elevation, the reactivation of latent viruses in an immune-compromised setting, neuroendocrine alteration involving changes in cortisol metabolism, and autonomic nervous system dysregulation [37]. Preliminary findings exploring genetic pathways in CRF provide support for further characterizing inflammatory pathways in the etiology and management of CRF.

### 2.2. Biobehavioral Risk Factors

Dietary behaviors are integrally tied to biobehavioral risk factors for CRF, including sleep and physical inactivity. Sleep disturbance (e.g., insomnia, sleep apnea, hypersomnia, parasomnias, and problems with circadian rhythm) is prevalent among cancer survivors with estimates ranging from 17–70% [51]. Sleep has a bi-directional relationship with diet and nutrition. In a randomized clinical trial in breast cancer survivors who completed cancer treatment, a three-month fatigue reduction diet characterized by high intakes of fruit, vegetables, whole grains, and omega-3 fatty acids improved fatigue (*p* = 0.01) and sleep (*p* = 0.03) compared with the general health curriculum group [52]. Recent research suggests lower levels of calcium, vitamin C, and selenium are associated with disrupted sleep [53], and lower overall diet quality has been associated with a shorter sleep duration [54]. There is also increasing evidence suggesting sleep has an influence on dietary choices. Both cross-sectional and epidemiologic studies have demonstrated that those who sleep less are more likely to consume energy-rich foods (e.g., refined carbohydrates and fats), fewer vegetables, and have irregular meal patterns [55]. In contrast, foods with high availability of tryptophan (e.g., milk, canned tuna, turkey, oats), a precursor to serotonin and melatonin, may be the most helpful in promoting sleep [55]. The exact mechanism linking sleep disturbance, diet, and CRF remains unclear, but preliminary evidence suggests abnormal cortisol levels, pro-inflammatory biomarkers of inflammation and angiogenesis, and circadian rhythm expression play a role [56,57]. The links among sleep disturbance, diet, and CRF suggest potential benefits of diet intervention as a non-toxic treatment strategy for CRF.

In addition to sleep, physical inactivity has been associated with increased CRF [58,59]. Inadequate dietary intake resulting in involuntary muscle wasting and muscle weakness contribute to CRF through altered ATP metabolism. ATP is a key mediator in generating new muscle tissue and assisting in muscle contractions. Thus, decreased ATP production may play a significant role in the development of CRF [59]. Vitamin D represents another potential link between physical activity and CRF. Vitamin D deficiency is prevalent among obese cancer survivors, and is associated with physical function, musculoskeletal pain, and alterations in body composition [60,61,62,63]. While further research on the mechanisms underlying biobehavioral relationships with CRF is needed, it is likely that biobehavioral risk factors have a synergistic effect on fatigue (e.g., sleep disruption contributes to fatigue, and physical inactivity further exacerbates fatigue).

## 3. Metabolomic Studies of Cancer-Related Fatigue

Studies identifying metabolic signatures of CRF are emerging across cancer populations, bolstered by an improvement in metabolomics technologies that measure up to thousands of metabolites from low volume biospecimens. Untargeted metabolomics investigations provide an unbiased means of discovering metabolites unique to CRF that could be modulated by dietary factors, thus unveiling potential dietary targets to intervene upon for mitigating or preventing CRF. Pathways of metabolic dysfunction that have been associated with increased fatigue include those consistent with a hypometabolic state (e.g., declines in sphingo- and glycosphingolipids, phospholipids, purines, microbiome aromatic amino acid, branch chain amino acids, and lathosterol) [64]. Numerous metabolic pathways have been associated with CRF in different cancer populations. Mechanisms that may lead to dysregulation include the regulation of ATP production and cellular energy, metabolism, and the steroid hormone biosynthesis pathway [65,66,67].

In a recent longitudinal study of cancer survivors with mixed cancer types, metabolites involved in sphingolipid, histidine, cysteine, and methionine metabolism were associated with increased CRF in cancer survivors as compared to cancer-free individuals [66]. A pilot study of post-treatment colorectal cancer survivors found that fatigue was associated with two metabolomic pathways involved in the regulation of ATP production and cellular energy (i.e., glutathione metabolism, and D-glutamine and D-glutamate metabolism) [67]. Fatigue worsens over time following androgen deprivation therapy (ADT) initiation, in a study of prostate cancer survivors receiving ADT, androstenedione, androgen, estrogen, amino acid, and glutathione metabolic pathways discriminated patients receiving ADT from those not receiving the treatment [65]. Among these metabolites, steroid hormone biosynthesis metabolites were most significantly correlated with fatigue severity [65]. A global metabolomics profiling of cerebrospinal fluid (CSF) in acute lymphoblastic leukemia (ALL) survivors identified eight metabolites associated with fatigue (*p* < 0.05). Fatigue was positively associated with dimethylglycine (amino acid pathway), allantoin (nucleotide pathway), ribitol (carbohydrate pathway), and dimethylmalonic acid (lipid pathway), while fatigue was negatively associated with gamma-glutamylglutamine (peptide pathway), 3-methoxytyrosine (amino acid pathway), asparagine (amino acid pathway), and myoinositol (lipid pathway) [68]. A pilot study of metabolite profiles among breast cancer survivors found that profiles among those with post-chemotherapy fatigue, compared with pre-chemotherapy fatigue, were characterized by higher concentrations of acetyl-l-alanine (amino acid) and indoxyl sulfate (indole) and lower levels of 5-oxo-l-proline (carboxylic acid) [69]. These findings could be useful for elucidating biological mechanisms associated with the development of pain and fatigue [69]. In a longitudinal study of hepatocellular carcinoma survivors, lenvatinib therapy affected the carnitine system by lowering carnitine levels, and carnitine insufficiency was associated with increased fatigue [70]. Thus, metabolites have been associated with fatigue across numerous cancer populations. Further research comparing metabolic signatures unique to cancer survivors with CRF vs. metabolites that may be associated with, but not unique to, CRF (i.e., overlap pathways) vs. metabolites associated with cancer but not CRF, will strengthen our understanding of metabolomic profiles specific to CRF [66] (Figure 1).

## 4. Dietary Recommendations for Cancer-Related Fatigue

Despite the burden of CRF among cancer survivors, few studies have investigated the role of diet in the development or management of CRF. Indeed, specific dietary guidelines for cancer survivors have been developed for some cancer types, but they focus on dietary relationships with disease recurrence or mortality rather than optimizing diet to prevent or manage cancer and treatment related side-effects [71]. Existing dietary interventions do not directly target CRF, focusing more indirectly on mechanisms tied to CRF, such as inflammation and lean muscle mass maintenance [31]. The few diet interventions provided prior to, during, and after cancer treatment that may influence CRF have previously been summarized [31]. Moreover, while various patient-reported survey measures for CRF are currently used, objective measures of CRF have not been validated. Identifying objective biomarkers of CRF would strengthen CRF outcome assessment in interventional studies [31].

Nutritional status plays an important role as a mediator of fatigue both directly and indirectly through its impact on underlying physiological mechanisms, such as poor nutritional status and micronutrient deficiencies [72]. Specific nutrition-related mechanisms posited to relate to CRF include chemotherapy-induced malnutrition, anemia, treatment-related nutrient malabsorption and altered metabolism (e.g., mitochondrial dysfunction), involuntary weight loss or gain, modification in body composition (e.g., obesity, undernutrition, and sarcopenia), and difficulty meeting protein-energy requirements in the presence of cancer and/or its treatment [72]. Fatigue is also thought to be a result of the dysphagia-malnutrition relationship [73], and chewing problems including loss of teeth further exacerbate malnutrition and fatigue [74,75]. Alterations in body composition due to cancer (e.g., muscle wasting-associated sarcopenia) or changes in dietary intake (e.g., undernutrition) [76] underscore the value of nutritional interventions targeted to counteract muscle decline in the context of fatigue [72].

Current dietary guidelines for cancer survivors recommend diets high in anti-inflammatory foods such as fruits, vegetables, whole grains, lean proteins, and healthy fats (specifically omega-3 and omega-6 fatty acids) and low in saturated fat and processed meat [77,78]. These dietary patterns after cancer diagnosis have been inversely associated with CRF [36,79,80,81,82,83,84]. In a prospective cohort study of colorectal cancer survivors, higher dietary fiber, fruit, and vegetable intakes were longitudinally associated with better physical functioning and less fatigue from six-weeks to 24 months post-treatment [85]. Macronutrients including omega-3 fatty acids, fiber, and protein, which assists in maintaining and building lean muscle mass, have all been associated with improvements in CRF randomized dietary trials [52,80]. For example, studies have shown lower fatigue among breast cancer survivors with higher dietary omega-3 fatty acid intake [52] and those with a higher fiber intake [80]. Additionally, high protein intake has been found to predict a lower CRF burden in advanced-stage cancer patients undergoing chemotherapy [79]. In contrast, in a review of nutritional interventions for CRF, a high intake of pro-inflammatory refined carbohydrates, including sugar and processed grains, has been associated with increased CRF burden [31]. Furthermore, caloric restriction (CR) of 30% without an incurrence of malnutrition was associated with decreased adiposity and inflammation, and improved insulin sensitivity, blood glucose, inflammation and angiogenesis in non-obese humans, suggesting a feasible intervention for reducing CRF [86]. However, chronic CR is contraindicated for many cancer patients at risk for weight loss, cachexia, and immunosuppression, thus intermittent CR or low carbohydrate/ketogenic diets may be more suitable [86].

More recently, the focus has shifted from individual food components to broader recommendations for overall dietary pattern. The World Cancer Research Fund/American Institute for Cancer Research (WCRF/AICR) diet and lifestyle guidelines now recommend a diet rich in whole grains, vegetables, fruits, and beans, with limited consumption of processed foods, red meat, sugar-sweetened drinks, and alcohol [87]. This dietary pattern largely aligns with the Dietary Guidelines for Americans, and other chronic disease-specific dietary guidelines [88,89,90]. Robust systematic reviews with dose-response meta-analyses contributed to the evidence-base supporting these guidelines. Dietary patterns associated with CRF, specifically, have been reviewed elsewhere. Briefly, anti-inflammatory dietary patterns (e.g., the Mediterranean diet [91,92], ketogenic diet [93,94], macrobiotic diet [80,95], Paleolithic diet [92], and other plant-based diets [80,96]) have been associated with reduced fatigue, possibly by lowering inflammation and improving gut microbial characteristics. Additionally, two recent studies [97,98] identified distinct gut microbial taxa associated with high compared with low fatigue among cancer survivors of multiple cancer types. Dietary patterns are one of the strongest factors influencing the gut microbiome [99]. The gut microbiome significantly influences immune response with the capacity to induce inflammation, which is relevant to the etiology of CRF [100]. Future studies that evaluate how the gut microbiome may mediate the relationship between diet and CRF are warranted.

Indeed, adherence to the WCRF/AICR diet and lifestyle guidelines has been associated with a decreased fatigue burden in several studies [101,102,103]. The alteration of dietary patterns after cancer diagnosis (i.e., increasing consumption of fruits, vegetables, fish, nuts and seeds) could be more important than total energy and/or protein intake prescriptions for reducing CRF [52]. A 2019 systematic review and meta-analysis of nutrition therapy for CRF and quality of life lent further support to the potential benefits of anti-inflammatory dietary patterns (i.e., high intake of fruits, vegetables, fish, nuts and seeds) in reducing CRF, however evidence to determine the optimal nutrition care plan to improve CRF and quality of life is insufficient [104]. A 2020 study among Malay breast cancer survivors found that an increased intake of green leafy and cruciferous vegetables and reduced intake of red meat, sugar, and fried cooking methods were associated with improved emotional function and reduced fatigue [105].

The WCRF/AICR diet and lifestyle guidelines recommend meeting nutritional needs through diet alone, and advise against using high-dose dietary supplements for cancer prevention. The guidelines advise cancer patients to follow the same recommendations as for cancer prevention. Dietary-supplement use in cancer survivors is common [106]; however, evidence supporting its beneficial health effects is limited. Only a handful of studies to date have examined associations between dietary supplement use and fatigue in cancer survivors, and these few studies show mixed results [107,108,109]. In a phase II nationwide study, among cancer survivors who supplemented with fish oil, greater increases in serum omega-3s were associated with greater improvements in fatigue [109]. Among colorectal cancer survivors using Vitamin B_6_ supplements, higher circulating pyridoxal 5′-phosphate (PLP), the active form of Vitamin B_6_, was cross-sectionally associated with reduced fatigue and improved physical and social functioning 6 months post-diagnosis [107]. However, a longitudinal study among colorectal cancer survivors showed no differences in fatigue between dietary supplement users (predominately multivitamins) and nonusers [108]. In contrast, the use of oral complementary and alternative medicine (OCAM), defined as taking homeopathy, vitamins/minerals, or herbal/dietary supplements, was associated with more severe CRF in stage I–III breast cancer survivors [110].

## 5. Challenges of Nutritional Metabolomics in Cancer-Related Fatigue

Nutritional interventions promoting dietary patterns that target CRF-relevant pathways such as inflammation provide a promising therapy for cancer survivors to prevent or manage CRF. However, few interventions have been conducted to date. Observational studies that evaluate dietary relationships with CRF remain limited by challenges of imprecise dietary intake assessment, and few cancer cohorts have implemented dietary assessments longitudinally after cancer diagnosis. Nutritional metabolomics has been employed to identify objective dietary biomarkers to reduce dietary measurement error and biases associated with dietary self-report [111,112,113,114,115,116,117,118,119]. Moreover, studies of nutritional metabolomics in relation to cancer risk and outcomes have provided objective validation of dietary relationships with cancer, and unveiled novel metabolic pathways linking diet to cancer [120].

Dietary intake is multi-dimensional and dynamic. Emerging evidence suggests that cancer survivors change their diet after diagnosis [121], particularly those experiencing physical symptoms [122]. Yet, few studies have characterized dietary intake across the cancer survivorship trajectory (pre-treatment, peri-treatment (0–6 months), early post treatment (6–24 months), and late-post treatment (24–36 months)) in conjunction with measuring cancer-relevant endpoints, including CRF. Future longitudinal studies will contribute immensely to the evidence base on diet intake history and relationships between CRF and dietary changes across survivorship. By leveraging validated food intake biomarkers, nutritional metabolomics offers a means of studying dietary changes among cancer survivors in the absence of self-reported questionnaires, which are difficult to implement in populations experiencing severe illness. Furthermore, nutritional metabolites identified in recent studies demonstrate greater specificity and more robust diet-biomarker associations than prior dietary biomarkers [123]. The utility of food biomarkers in a dietary interventional setting extends as a tool for monitoring compliance to dietary intervention, which is a recognized challenge. Yet, their use in measuring diet in cancer survivor populations is limited.

Nutritional interventions for cancer survivors with CRF represent a new area of research. Given the application of nutritional metabolomics to better understand the role of diet in cancer risk [120], a similar application could be used to understand the role of diet in the etiology and/or management of CRF. Untargeted metabolomics (i.e., identifying all measurable metabolites in a sample) with machine learning approaches could be used to identify groups of metabolites that distinguish CRF from patients without fatigue, and subsequent analyses could identify food combinations or dietary patterns associated with those metabolic profiles. In a similar vein, endogenous metabolites associated with CRF dietary patterns could unveil potential metabolic mediators. Secondly, a priori-defined food biomarkers that have reached an appropriate level of validation [124] could be measured in association with CRF to test specific diet exposure-CRF hypotheses. For example, biomarkers of coffee (quinate, 3-hydroxypyridine sulfate, 1,3-dimethylurate), alcohol (ethyl glucuronide), multivitamins (pantothenate (B_5_), pyridoxal, alpha-tocopherol) and citrus fruits (proline betaine) have been validated in large-scale cohort studies and subsequently tested in acute dietary intervention or feeding studies [116,125,126,127,128,129,130,131]. Leveraging nutritional metabolomics data derived from other types of biospecimens (e.g., fecal metabolites) may also provide novel insights, such as diet-microbiome interaction in relation to CRF.

Dietary pattern analysis is increasingly recognized as an important method to account for the totality of diet, the complexity of its influence on biochemical pathways, and, ultimately, on health and symptom development [132]. Pre-defined indices of diet quality have been associated with metabolites in several studies, underscoring the endogenous pathways possibly influenced by a certain pattern of eating and, and the specific dietary index components (i.e., foods) that may drive associations with disease outcomes (e.g., the Healthy Eating Index-2015 (HEI-2015), scoring adherence to the Dietary Guidelines for Americans [133,134,135] and the Alternative Mediterranean Diet (aMED) [136,137,138,139]). Empirical, hypothesis-oriented dietary patterns are developed using reduced rank regression and dimensionality reduction approaches. They are characterized by groups of specific foods that, when consumed together, have the strongest association with specific biomarkers. Such dietary patterns have been developed and validated in different populations to explain variations in inflammatory and insulinemic biomarkers [35,140]. A similar approach could be used to develop an empirical dietary index for CRF, following the identification of validated CRF biomarkers [35,140,141,142]. A next step would be testing such a dietary pattern in dietary interventions for the prevention or management of CRF in cancer patients.

## 6. Conclusions

CRF is prevalent and debilitating for cancer survivors. In the last decade, there has been a surge in research focused on identifying CRF risk factors and potential interventions to target these etiological factors [143]. Studies examining links between CRF and dietary inflammation largely support the hypothesis that inflammatory processes contribute to fatigue during and after cancer treatment [5]. Currently, our understanding and management of CRF is limited by unclear etiological mechanisms, a dearth of evidence-based guidelines for the nutritional management of CRF, and weak epidemiologic evidence for associations between CRF and dietary factors. The application of untargeted nutritional metabolomics with machine learning approaches could identify groups of metabolites that distinguish survivors with CRF from survivors without fatigue. Subsequent analytic approaches could identify food combinations and dietary patterns associated with unique CRF metabolic profiles, thus informing dietary intervention research to specifically target CRF. Harnessing metabolomics to examine the role of dietary biomarkers in CRF could facilitate an objective dietary assessment, mitigate imprecision related to self-reported diet, and identify mechanisms underlying the association of diet with CRF. A nutritional metabolomics approach to understanding the etiology of CRF could in turn strengthen evidence-based dietary guidelines for cancer survivors and optimize CRF interventions to reduce symptom burden.

## Figures and Tables

**Figure 1 nutrients-14-01496-f001:**
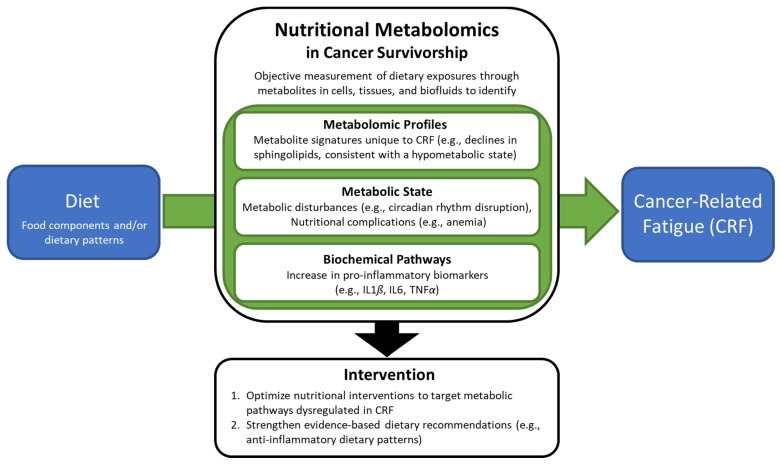
Nutritional Metabolomics in Cancer Survivorship.

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
