# Peer review of "A Molecular Approach to Understanding the Role of Diet in Cancer-Related Fatigue: Challenges and Future Opportunities"

_nutrients, 2022, doi:10.3390/nu14071496_

Round 1
Reviewer 1 Report
This paper seems to discuss CRF in cancer survivors. This should be shown more clearly and it may be worthwhile to concentrate the issues to those for cancer survivors. For this purpose, it may be helpful to clarify whether the cited literature and evidence are for the cancer survivors or not.
I think it better understandable not to discuss cancer survivors at the same time with patients who are undergoing chemotherapy for advanced cancer and/or cachexic advance cancer patients.
CRF seems to be a complex syndrome and treatment strategies of CRF may be different depending on the patient’s clinical status; ie patients are under active cancer treatment or posttreatment without active treatment or end of life. For example, corticosteroids may provide short-term relief for fatigue and improve QOL in terminal cancer patients however may not be suitable for long term use in posttreatment cancer survivors. Diet may play important roles especially in post treatment phase.
It seems that this paper focuses mainly on cancer survivors (post treatment without active treatment) and this I think is of the most interest because fatigue has been shown to persist into long periods of posttreatment. But the cited literature and evidence are being discussed without distinction.
Just as a few examples, meta-analysis cited (ref 2) include patients other than long term cancer survivors and pharmacologic intervention studies include patients under active cancer treatments. There are some statements on circulating cytokines. It is well known that cytokines are important factor for the development of many of the cancer related (associated) symptoms including anorexia, cachexia and weight loss. It is plausible that circulating cytokines may take some roles in the CRF. I would like to know if there are any of these data available on cancer survivors without active treatment.
I think it better understandable to clarify points outlined above thought the manuscript.
Author Response
Reviewer 1 Comments:
- This paper seems to discuss CRF in cancer survivors. This should be shown more clearly and it may be worthwhile to concentrate the issues to those for cancer survivors. For this purpose, it may be helpful to clarify whether the cited literature and evidence are for the cancer survivors or not.
Response: We thank the reviewer for this comment. As long-term maintenance and hormonal therapies now blur the line for many patients regarding the end of treatment (e.g., chemotherapy) as it has traditionally been considered, we have updated the introduction section of the manuscript to include the American Cancer Society definition for cancer survivors.
Page 1, Line 32-38: The American Cancer Society defines a cancer survivor as “anyone who has ever been diagnosed with cancer no matter where they are in the course of their disease [1].” For many survivors, cancer-related fatigue (CRF) is a “distressing, persistent, subjective sense of physical, emotional, and/or cognitive tiredness or exhaustion related to cancer or cancer treatment that is not proportional to recent activity and interferes with usual functioning” [2], and prevention and interventional strategies are needed to address this growing challenge.
- I think it better understandable not to discuss cancer survivors at the same time with patients who are undergoing chemotherapy for advanced cancer and/or cachexic advance cancer patients.
Response: We thank the reviewer for this comment and have updated the introduction section of the manuscript to include the American Cancer Society definition for cancer survivors that states a survivor is anyone who has ever been diagnosed with cancer no matter where they are in the course of their disease. As cancer-related fatigue is a concern for survivors before, during, and after treatment it may be important to discuss cancer-related fatigue throughout the disease continuum.
Page 1, Line 43-44: A recent systematic review suggests that over half of all cancer survivors experience CRF, [3] and 96% of patients undergoing chemotherapy experience CRF [9].
- CRF seems to be a complex syndrome and treatment strategies of CRF may be different depending on the patient’s clinical status; (i.e., patients are under active cancer treatment or posttreatment without active treatment or end of life). For example, corticosteroids may provide short-term relief for fatigue and improve QOL in terminal cancer patients however may not be suitable for long term use in posttreatment cancer survivors. Diet may play important roles especially in post treatment phase.
Response: This is an excellent point, and this text has been added to the manuscript.
Page 2, Line 57-65. CRF is a complex syndrome, and treatment strategies of CRF may differ depending on the survivor’s clinical status (e.g., under active treatment, post-treatment, end-of-life care). While corticosteroids may provide short-term relief for fatigue and improve quality of life in terminal patients, they may not be suitable for long term use in post-treatment survivors [30]. Additional treatment options for CRF are clearly needed. Nutritional interventions are a promising strategy for reducing CRF [31], but research is needed to elucidate the potential contributions of diet to CRF development and severity to isolate dietary targets that could help mitigate its devastating impacts.
- It seems that this paper focuses mainly on cancer survivors (post treatment without active treatment) and this I think is of the most interest because fatigue has been shown to persist into long periods of posttreatment. But the cited literature and evidence are being discussed without distinction. Just as a few examples, meta-analysis cited (ref 2) include patients other than long term cancer survivors and pharmacologic intervention studies include patients under active cancer treatments. There are some statements on circulating cytokines. It is well known that cytokines are important factor for the development of many of the cancer related (associated) symptoms including anorexia, cachexia and weight loss. It is plausible that circulating cytokines may take some roles in the CRF. I would like to know if there are any of these data available on cancer survivors without active treatment.
Response: We thank the reviewer for this comment and have updated the introduction section of the manuscript to include the American Cancer Society definition for cancer survivors that defines a survivor as anyone who has ever been diagnosed with cancer no matter where they are in the course of their disease. The authors acknowledge that cancer-related fatigue is a concern for survivors before, during, and after treatment and believe it is important to discuss cancer-related fatigue throughout the disease continuum. The authors have added clarifying terminology to the cited articles to decipher between active and post-treatment cancer survivors. The authors have also added language to the manuscript to specify discussion of cancer-related fatigue and circulating cytokines among post-treatment cancer survivors.
Page 3, Line 101-104: In a study of long-term survivors of testicular cancer, higher levels of circulating interleukin-1 receptor antagonist (IL-1ra) and C-Reactive protein (CRP) were found in those survivors with chronic CRF, thus lending support to the hypothesis that low-grade inflammatory processed are involved in chronic CRF [43].
Reviewer 2 Report
- The manuscript lack of enough references to prove their point of view.
- the conclusion of the manuscript could not clearly identify the role of molecular approach on studying diet in CRF.
- the illustration and argument of the article were too vague.
- lack of tables and fiures to describe the correlation of dietary supplements, nutrients and CRF.
Author Response
Review 2 Comments:
- The manuscript lack of enough references to prove their point of view.
Response: We thank the reviewers for this comment. We have cited over 140 references in the manuscript and have done an extensive search with experts in the metabolomics and cancer-related fatigue field. Unfortunately, at this time, few studies have explored the role in nutritional metabolomics in cancer-related fatigue, and we have highlighted this concerns in the dietary recommendations for cancer-related fatigue section.
Page 5, Line 203-209: Existing dietary interventions do not directly target CRF, focusing more indirectly on mechanisms tied to CRF, such as inflammation and lean muscle mass maintenance [29]. The few diet interventions provided prior to, during, and after cancer treatment that may influence CRF have previously been summarized [29]. Moreover, while various patient-reported survey measures for CRF are currently used, objective measures of CRF have not been validated. Identifying objective biomarkers of CRF would strengthen CRF outcome assessment in interventional studies [29].
- The conclusion of the manuscript could not clearly identify the role of molecular approach on studying diet in CRF.
Response: The authors have edited the conclusion to further clarify the role of metabolomics in studying diet in cancer-related fatigue.
Page 8, Line 362-366: The application of untargeted nutritional metabolomics with machine learning approaches could identify groups of metabolites that distinguish survivors with CRF from survivors without fatigue. Subsequent analytic approaches could identify food combinations and dietary patterns associated with unique CRF metabolic profiles, thus informing dietary intervention research to specifically target CRF.
- The illustration and argument of the article were too vague.
Response: We have updated the title of the manuscript to “A Molecular Approach to Understanding the Role of Diet in Cancer-Related Fatigue: Challenges and Future Opportunities.” Figure 1 has been updated to provide specific examples of nutritional metabolomics. A graphical abstract has been added to the manuscript to further summarize the correlations of dietary supplements, nutrients, and cancer-related fatigue.
- Lack of tables and figures to describe the correlation of dietary supplements, nutrients and CRF.
Response: Figure 1 has been updated to provide specific examples of nutritional metabolomics. A graphical abstract has been added to the manuscript to further summarize the correlations of dietary supplements, nutrients, and cancer-related fatigue.
Round 2
Reviewer 1 Report
I think the manuscript has been improved.